# Extracting Statistical Properties of Solar and Photovoltaic Power Production for the Scope of Building a Sophisticated Forecasting Framework

**Joseph Ndong [1,*] and Ted Soubdhan [2]**

1   Faculty of Sciences and Techniques, Department of Mathematics and Computer Science, University Cheikh Anta Diop of Dakar, Dakar 10700, Senegal
2   Laboratoire LARGE, Faculty of Sciences, Department of Physics, University of Antilles, 97157 Pointe-à-Pitre, France; ted.soubdhan@univ-antilles.fr
*   Correspondence: joseph.ndong@ucad.edu.sn

**Abstract:** Building a sophisticated forecasting framework for solar and photovoltaic power production in geographic zones with severe meteorological conditions is very challenging. This difficulty is linked to the high variability of the global solar radiation on which the energy production depends. A suitable forecasting framework might take into account this high variability and could be able to adjust/re-adjust model parameters to reduce sensitivity to estimation errors. The framework should also be able to re-adapt the model parameters whenever the atmospheric conditions change drastically or suddenly—this changes according to microscopic variations. This work presents a new methodology to analyze carefully the meaningful features of global solar radiation variability and extract some relevant information about the probabilistic laws which governs its dynamic evolution. The work establishes a framework able to identify the macroscopic variations from the solar irradiance. The different categories of variability correspond to different levels of meteorological conditions and events and can occur in different time intervals. Thereafter, the tool will be able to extract the abrupt changes, corresponding to microscopic variations, inside each level of variability. The methodology is based on a combination of probability and possibility theory. An unsupervised clustering technique based on a Gaussian mixture model is proposed to identify, first, the categories of variability and, using a hidden Markov model, we study the temporal dependency of the process to identify the dynamic evolution of the solar irradiance as different temporal states. Finally, by means of some transformations of probabilities to possibilities, we identify the abrupt changes in the solar radiation. The study is performed in Guadeloupe, where we have a long record of global solar radiation data recorded at 1 Hertz.

**Keywords:** bayesian inference; GMM; HMM; viterbi decoder; possibility theory

## 1. Introduction

The major difficulty in setting up a statistical prediction tool for PV power production is to take into account fluctuating atmospheric conditions. Photovoltaïc (PV) energy production depends strongly on solar irradiance which depends on meteorological conditions. A significant challenge related to building an efficient PV management system is to perform suitable prediction for PV production. Due to the important problem related to the high variability observed in the solar irradiance, since the daily atmospheric conditions (weather, temperature, etc.) encounter many changes, this challenge can be difficult, especially under tropical conditions. The variability in solar resources poses difficulties in grid management, as solar penetration rates rise continuously. In addition, fluctuations in solar irradiance can generate long-memory dependencies leading to the non-stationarity of the underlying process. Therefore, forecast methods generally lack accuracy because they cannot capture these long-term dependencies. In order to generate reliable future estimation of PV production,

it is important to build a suitable forecasting tool able to automatically adjust the model parameters when the weather conditions encounter changes. Several works in the field of solar energy prediction make the strong assumption that the entire process is stationary. However, under the conditions of untimely changes in atmospheric conditions, as can be observed under tropical climate, this assertion may suffer from some shortcomings, thus leading to modeling errors.

To the best of our current knowledge, the literature does not provide physical and statistical forecasting tools capable of automatically selecting the best PV forecasting model from a given set of candidate models, in accordance with the dynamic changes caused by atmospheric conditions. The literature is full of very good papers related to solar and PV power-production forecasts [1–3]. These works, which focus on statistical prediction methods, use very powerful machine-learning techniques such as SVM (support vector machine), PCA (principal component analysis), ARMA, Kalman filter, and neural networks, etc. However, they do not solve the delicate problem of significant fluctuations in solar irradiance and sudden changes caused by weather conditions [4]. In addition, the proposed models are generally based on the calibration of a single process and, very often, this requires a lot of data to improve the accuracy of the results. None of the proposed approaches offers a system in which we have the possibility of switching from one model to another taking into account changes in the dynamics of solar irradiance. In addition, several methods make strong assumptions about stationarity in order to be able to apply machine-learning techniques. Such an assertion can lead to significant prediction errors if the model parameters are not well-calibrated. In [5], we created a prediction tool which is based on the assumption of the stationarity of the global process. In addition, we obtained results with an RMSE around 10%, which we can consider to be non-negligible and which might be improved. The calibration of the model parameters was performed carefully by a maximization algorithm, to minimize prediction errors.

On the other hand, the issue related to the fluctuations in solar irradiance in the field of the implementation of tools for forecasting solar and photovoltaic PV power has attracted the interest of utilities and researchers towards developing state-of-the-art forecasting techniques for forecasting wind speeds and solar irradiance over a wide range of temporal and spatial horizons. Several works have been published to try to understand how to predict solar irradiance in order to set up efficient forecast frameworks. One can find a good review of the solar-irradiance forecasting techniques in [6,7]. The main forecasting approaches employ physical, statistical, artificial intelligence and hybrid methodologies. The main drawback in these works is that most of the proposed methods require a huge volume of data, which can increase the complexity requirements. Only the so-called 'persistence model' technique uses a small volume of data. These techniques also suffer from the potential problem of correlation between wind and solar irradiance. The accuracy of the predictions over different horizons is severely impacted if the spatio-temporal correlations are not well-identified. The most striking thing about these works is that, no method has given the possibility of discovering, for the different time horizons, both the classes of variability and, in each of them, the sudden and untimely, or abrupt, changes in atmospheric conditions.

Therefore, our aim consists of providing a first case study for the need of developing an effective tool to predict solar irradiance, in the sense that the system is composed of a set of several statistical models/classes. In addition, the management system will be able to automatically select the appropriate model according to the weather conditions and cloud distribution. We believe that it is possible to build a prediction model much less sensitive to errors. To achieve this goal, the model must be robust enough to identify and extract meaningful information (statistically) about the dynamic evolution of the solar radiation to achieve good performance. We believe that the different levels of solar-irradiance variability can be captured in different models, each model having its own calibration parameters. Consequently, it would be possible to see the entire model as a series of parametric models and the challenge will become being able to automatically select and calibrate dynamically

the appropriate model whenever the corresponding atmospheric conditions exist. For this reason, in this paper, we implement a robust monitoring framework for a thorough analysis of the dynamics of solar irradiance to make a tool robust enough to discover both the macroscopic variations seen as the different levels of solar-radiation amplitude and the underlying microscopic variations or abrupt changes. For this scope, we build a methodology based on combining statistical inference techniques and possibility theory. Statistical inference methods based on a Gaussian mixture modeller and a hidden Markov model can be properly used to extract the different levels of variability in the global solar-radiation data. In addition, finally, we use a convenient method related to combining probability and possibility theories to learn more about the intrinsic variability of each class of variability. In this work, we use global solar-radiation data recorded with a time step of one second over three years, from 2011 to 2013, at the facility of the LARGE laboratory.

In summary, the solar-irradiance prediction framework that we propose allows :

1     The detection of the different MAVs corresponding to solar classes of variability;
2     The identification of MIVs (abrupt changes) inside each MAV;
3     The study of the probabilistic characteristics of each MAV to build a means of determining whether or not the framework should renew the MAVs.

## 2. Methodology

The aim of this work is to build a sophisticated framework for solar resources forecasting, in real-time conditions. We believe that to achieve high accuracy for this purpose, this framework should take into account the dynamic evolution of the variability in the solar irradiance, which might encounter many abrupt changes. The high variability observed in the meteorological conditions (clear sky, cloudy sky, wind, and temperature, etc.) are the major sources of the different levels of variation for the measured irradiance. The main levels of variability in solar irradiance can be classified into two categories. The first category, named "macroscopic variations" (MAV), corresponds to a time interval $[t, t + \delta t]$ where the conditions remained stable and where the sky is either clear or cloudy, making it possible to observe the same production of solar irradiance. The second kind of variability consists of "microscopic variations" (MIV) inside the MAV. An MIV corresponds to points of time or short time intervals in $[t, t + \delta t]$ inside a given MAV and when sudden changes in meteorological conditions arise causing abrupt changes in solar fluctuations. Abrupt changes can be seen as points of time where the irradiance encounters an important increasing or decreasing in its amplitude. In order to extract the MAVs and, consequently, the MIVs, we follow a methodology which applies a series of mathematical techniques, as shown in the architecture described in Figure 1.

As a first step, we assume that the real distribution of the process representing the solar radiation is an ensemble of normally distributed processes. This assertion allows us to build a method which has the ability to model the MAVs with different families of Gaussians. Therefore, with the calibration of a Gaussian mixture model (GMM), we find the K best number of clusters corresponding to the MAVs. Thereafter, since we plan to study the dynamic evolution of the variability over time and to detect the abrupt changes or MIVs, we follow a second step where we calibrate a hidden Markov model by means of the Viterbi decoder to learn the temporal dependencies of the detected MAVs. At this point, we can see how the different MAVs evolve over time intervals.

We refer the reader to the papers [8–11] to gain a complete view of the calibration of the GMM and HMM models.

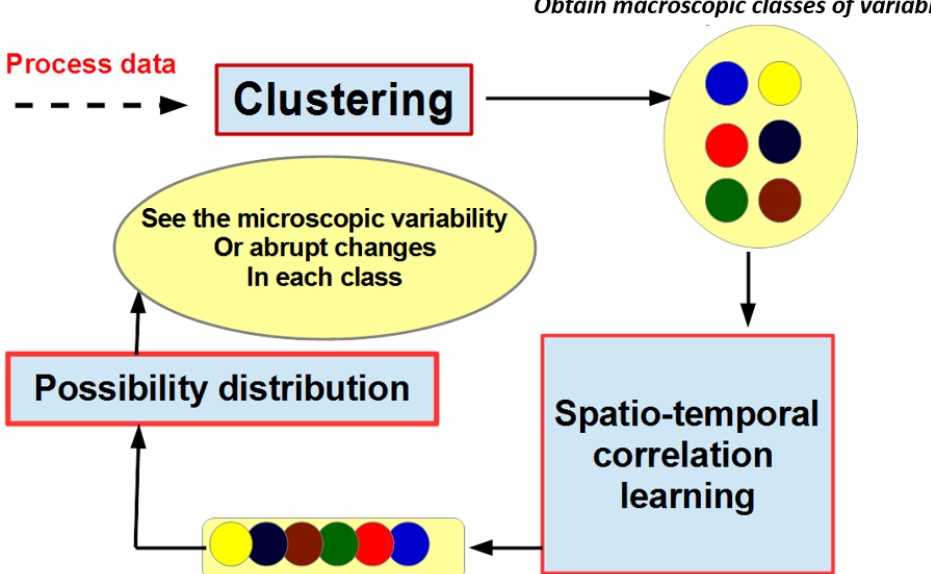

**Figure 1.** Architecture to track the dynamic evolution of the variability in solar irradiance. We show how the different mathematical tools are linked to achieve the discovery of the different levels of solar-irradiance variability and the underlying microscopic changes.

Recall that our main objective for this present work is to detect both the MAVs and, inside each MAV, the MIVs. The technique we developed to identify MIVs is based on possibility theory. Probability theory has a long and successful record for solving the problem of uncertainly. Possibility theory is another innovative field which can go beyond the study of uncertainly by taking into account the incompleteness and inconsistency of the data. This new approach gives us a sophisticated tool to solve the generally hard problem of building thresholds in order to separate given processes into several sub-spaces.

## 3. Organization of the Paper

In Section 4, we discuss previous works related to forecasting methods for solar irradiance and PV power production. We will see that current research does not offer a framework which detects both macro and micro variations and the possibility of automatic model selection based on probability laws. The detection of MAV variables is performed using Bayesian inference tools using probabilistic models based on Gaussian distribution laws. Section 5, therefore, uses the approaches of GMM and HMM models to determine the classes of variability (MAVs) and study the spatio-temporal correlations in order to locate the latter in the temporal scale. In this section, we also show how to calibrate and turn the two model parameters. In Section 6, we discuss how the detection of MIVs can be performed by a thorough use of the paradigm of possibility theory. Section 6.1 details the implementation details for possibility theory which can be used to effectively quantify the degree of possibility of each MAV. Section 6.2 focuses on the implementation details to quantify each MIV by a degree of possibility. Section 7 discusses the validaton of the whole procedure and the results obtained. Section 8 is reserved for the explanation of how to use the results of the analysis of the solar irradiance to set up a tool for predicting the production of photovoltaic energy. Section 9 gives som challenges in mastering the proper implementation of such a framework and in Section 10 we give conclusions and future perspectives.

## 4. Related Works

We are interested in investigating the field of solar- and photovoltaic-power production forecasting techniques. We focus on case studies where the energy exploitation zone

undergoes strong meteorological changes. In this case, it would be judicious to provide a framework in which the management infrastructure can automatically choose a prediction model depending on weather conditions. According to current knowledge, there are no tools for this purpose. However, several works relating to the prediction of global solar irradiance are proposed in the literature, but none of them develops a tool which, at the same time, simultaneously detects the changes which we have defined as MAV and MIV [1–3]. The different statistical approaches used apply techniques which analyze a single process describing the dynamics of the system. Authors often make assumptions that allow system parameters to be calibrated only once, which may result in larger prediction errors. We recently proposed a robust PV forecasting technique based on a linear dynamical Kalman filter [5]. The calibration of the proposed model can be performed either by an autoregressive model or by the EM (expectation-maximization) algorithm [12,13]. The method, also, was able to take into account exogenous variables (temperature, cloud cover, etc.) and we obtained better results than without these features. Compared to others techniques in the literature, the approach gave a better performance, with a RMSE of around 10%. We made the strong assumption of accepting that the process under consideration is stationary, which has the effect that the calibration of the model was performed only once, at the beginning (i.e., the model parameters were calculated once). A method of process normalization was proposed to make the assumption reliable. The data used to validate the model were collected in the sub-tropical zone of Guadeloupe where the atmospheric conditions, more often, change drastically in very short time intervals, thus having a great influence on the solar irradiation on which the PV production depends [14,15]. Therefore, we think that, in this situation, the process which governs the dynamics of the solar irradiance might be composed of many components with different statistical properties. If this assumption is true, it would be necessary to define a multi-model forecasting framework to suitably analyze the entire system. Thus, beyond this study, we believe that the error rate can be reduced to a much lower level if we analyze the high variability in the solar flux, which has a significant impact on energy production. The different levels of this variability might correspond to different probabilistic models.

We believe that all the proposed methods could provide better results if they were able to predict even the type of model that would be most accurate in predicting production. Therefore, this work is dedicated to studying and analyzing the features of the solar irradiance variability with the ultimate aim of building more sophisticated PV forecasting methods. To the best of our knowledge, this study is the first of a series of works that we will complete to arrive at such solutions.

## 5. Tracking the Macroscopic Variabilities-MAV

In this work, we use mathematical tools to achieve our hope of building a technique for solar-power variability detection. We plane to mathematically model the process governing the evolution of this feature by a finite set of probability distributions. We propose the usage of a parametric model to capture the dynamics of the variations by a family of probabilistic distribution functions. We assume that the underlying distribution of each type of variability can be modeled by a Gaussian random process and so the entire evolution of the system can be modeled by a mixture of Gaussians. Mixture distributions, in particular normal mixtures, are applied to data for two main purposes. One is to provide a semiparametric framework in which to model unknown distributional shapes, as an alternative to the kernel density method. The other is to use the mixture model to provide a probabilistic clustering of the data into g clusters corresponding to the g components in the mixture model. Anvari et al. [16] provide strong evidence that renewable wind and solar sources exhibit multiple types of variability and non-linearity in the timescale of seconds. Wind-power outputs for six hours can be forecasted by Gaussian processes in order to reflect the trend of wind power in the optimization framework by Lee et al. [17]. Therefore, we propose to explore the latter purpose of a mixture model, since we believe that the different levels of variability in solar power is an ensemble of Gaussian distributions.

We follow a three-phase process to run the the tool in order to detect MAVs. First of all, since we believe that the underlying distribution of a type of variation is Gaussian, we can retrieve, from the entire dataset, all the levels of variability. A tool based on Gaussian mixture model (GMM) is suitable to achieve this aim. This first step consists of an unsupervised classification operation which makes possible only the discovery of the classes of variability or MAVs. The only challenging task here is to identify the best number of classes. Afterwards, we wish to study the dynamic evolution of these classes on the time scale in order to see how these are linked on the temporal horizon. This second step consists of studying the temporal dependencies of the classes of variability. To solve this issue, we propose the use of a hidden Markov model (HMM), which gives, as output, a set of states, each of them containing a subset of the Gaussian components. The forward-backward algorithm helps us to achieve this second step. However, this step shows that, in reality, there are several possible sequences of states (inside the HMM trellis) which capture the dynamic evolution of the variability classes on the time scale. In addition, we would like to know which sequence of states in this HMM trellis is the unique best-state sequence, which explains the dynamic evolution of the process over time. Therefore, as a third and final step, we follow the forward-backward algorithm to another algorithm, namely, the Viterbi decoder, to extract the single best sequence of states evolving over time. In other words, the Viterbi algorithm is suitable to identify the true temporal dependencies of the underlying processes of the identified states.

*5.1. Model Selection Criteria for GMM*

The main challenge to overcome in order to ensure that the tool performs well is certainly the calibration of the GMM model. Therefore, the main question is how many components to include in the normal mixture model. If the ideal number of components of the GMM is found, this will ensure, in part, that the HMM will be well-calibrated, since the number of states is equal or less than the number of Gaussian components. The beginning of our dataset is used as learning data. By applying the EM algorithm [18], we calibrated a set of $g$ GMM models ($g = \{2, 3, 4, \dots, \}$ classes) and put our choice in the model, which minimizes some criteria.

A Gaussian mixture model (GMM) is a probabilistic framework which builds clusters based on a family of Gaussian/normal densities. GMMs are well-known for their ability to represent arbitrarily complex distributions with multiple modes. A GMM is based on density estimation with a linear combination of component densities of the form:

$$p(x) = \sum_{j=1}^{M} p(x|j)P(j), \tag{1}$$

where $p(x|j)$ are the Gaussian component densities and $P(j)$ the mixing parameters. To build a GMM, one has to choose a mixture component based on $P(j)$ and generate a data point $x$ from the chosen component using $p(x|j)$. This framework is generally used to perform unsupervised clustering, since the clusters are a-priori unknown (they constitute the hidden states of the mixture model).

It has already been stated that a mixture density with $g$ components might be empirically indistinguishable from one with either fewer than $g$ components or more than $r$ components, McLachlan et al. [19,20]. It is, therefore, sensible in practice to approach the question of the number of components in a mixture model in terms of an assessment of the smallest number of components in the mixture compatible with the data. Therefore, the true order $g_o$ of the $g$-component normal mixture model is defined to be the smallest value of $g$ such that the model is compatible with the data, with the model having different normal components and with all the associated mixing proportions $\pi_m$ being non-zero. There is a large number of criteria for selecting the ideal number of components, but here we focus on two criteria, namely, AIC (Akaike's information criterion) and BIC (Bayesian information criterion), McLachlan et al. [20].

To derive this two quantities, let us rewrite equation Equation (1) as follow:

$$f(x, \Psi_g) = \sum_{i=1}^{g} \pi_i \phi(x; \mu_i, \Sigma_i), \qquad (2)$$

The vector $\Psi_g$ consists of the mixing proportions $\pi_i$, the elements of the components means $\mu_i$ and the elements of the component covariance matrices $\Sigma_i$. The unknown parameters $\Psi_g$ are estimated by maximum likelihood via the EM algorithm [18]. AIC selects the best number of clusters by minimizing the quantity, McLachlan et al. [20]:

$$-2\log L(\hat{\Psi}_g) + 2d; \qquad (3)$$

where the log-likelihood is maximized over the model parameters and the penalty terms involve $d$, which denotes the total number of parameters in the $g$ model. For more information on the AIC criterion, see Akaike [21,22].

On the other hand, the BIC criterion of Schwarz [23] gives the ideal number of components of the model by minimizing the quantity:

$$-2\log L(\hat{\Psi}_g) + d\log n; \qquad (4)$$

In addition to the model selection purpose, one should know that each successive EM iteration will not decrease the likelihood, a property not shared by most other gradient-based maximization techniques. Therefore, if one runs the algorithm for a model $g$ and discovers that the likelihood decreases for successive iterations, he/she will conclude, in this case, that it would not be necessary to analyze these two criteria in order to know if the model in question is suitable.

*5.2. Initial Parameter Setting and Tuning for the HMM*

The calibration of an HMM is not straightforward, since the initialization procedure is a crucial phase to ensure that the model achieves good performance. Generally, this initialization consists of performing an initial random guess using a short part of the learning dataset. Instead, we use a more elaborated technique related to using the outputs of the Gaussian mixture modeler as input to set up the initialization parameters of the HMM. The HMM requires a transition matrix between states. To find this matrix, we apply a maximum a-posteriori (MAP) criterion to the classes of GMM in order to discretize the data (in symbols 1, 2, 3, etc.). Thereafter, for each symbol of this set, we simply calculate its proportion in the set with respect to the other symbols and we construct our transition matrix ($N \times P$), where N is the number of states and P the number of symbols or clusters. The mixing probability matrix (observation matrix) is simply obtained by calculating for each symbol its proportion with respect to the overall learning data.

*5.3. Results of the Statistical Properties of the HMM States*

After applying the Viterbi decoder, we discover the single best-state sequence suitable to capture the dynamic evolution over time of the variability in the solar-power irradiance. In addition to the analysis of the accuracy of the detection of the different levels of variability, we can show that the corresponding states are statistically different. To perform this accuracy measure, we calculate three quantities, namely, the mean ($\mu$), variance ($\sigma$) and coefficient of variation ($\frac{\sigma}{\mu}$) on suitably well-defined sequences of data for each state. To this end, we progress in three steps. First, after applying the Viterbi algorithm, for each obtained state, we extract the largest continuous sub-sequence. Let us say that we construct the vector $T$ containing all $T_s$, where $T_s$ is the length of the largest sub-sequence in state $s$. This largest sub-sequence corresponds to the highest time period where a level of variability occurs for that state. Afterwards, we take the maximum of $T$ (let us denote $T_{max}$) and then the whole process of the three combined states is subdivided into several sub-sequences of size $T_{max}$. Finally, we calculate, for each sub-sequence, the mean, variance and coefficient of variation

to see the accuracy of the detection and separation of the different levels of variability by showing that their respective statistics are typically singular. The coefficient of variation, which is the ratio between the variance and the mean, measures the dispersion of the data around the mean because the standard deviation alone often makes it impossible to gauge the dispersion. This statistic serves to perform comparison between many data series.

## 6. Possibility Theory as a Tool to Detect MIVs Inside an MAV

We present here the details of a new formalism based on possibility theory to analyze the MAVs of the solar-irradiance variability and extract the abrupt changes defined as microscopic variations (MIVs). Afterwards, we will see how to extract the probabilistic laws of the different types of variations. We follow a two-step process to do so. First, we need to associate to each cluster/MAV a degree of possibility to quantify its existence inside the overall process and, as a second step, we quantify each occurrence of data inside an MAV by a degree of possibility. An MIV will correspond to a time instant where the degree of possibility of the corresponding data has a certain value.

A consistency principle between probability and possibility can be stated in a non-formal way [24]: "*what is probable should be possible*". This requirement can then be translated via the inequality:

$$P(A) \leq \Pi(A) \qquad \forall A \subseteq \Omega \tag{5}$$

where $P$ and $\Pi$ are, respectively, a probability and a possibility measure on the domain $\Omega$. In this case, $\Pi$ is said to **dominate** $P$. This consistency principle means that we can always use possibility theory to build a more reliable statistical test by combining probability and possibility, as we can see in the following.

Dubois and Prade built a procedure [25,26] which produces the most specific possibility distribution among the ones dominating a given probability distribution. In this paper, this method is generalized to the case where the prior probabilities (of generating the GMM clusters/MAVs) are **unknown**. We assume the above clusters were generated from an unknown probability distribution. It is proposed to characterize the probabilities of generating the different clusters by *simultaneous confidence intervals* with a given confidence level $1 - \alpha$. A procedure for constructing a possibility distribution is described, insuring that the resulting possibility distribution will dominate the true probability distribution in at least $100(1 - \alpha)$ of the cases.

### 6.1. Inferring Possibility Distribution for the MAVs

Let $n_k$ denote the number of occurrences of cluster $k$ in a sample of size $N$. Then, the random vector $n = (n_1, \ldots, n_K)$ can be considered as a *multinomial* distribution with parameter $p = (p_1, p_2, \ldots, p_K)$. A confidence region for $p$ at level $1 - \alpha$ can be computed using *simultaneous confidence intervals*, as described in [27]. Such a confidence region can be considered as a set of probability distributions.

It is proposed to characterize the probabilities $p = (p_1, p_2, \ldots, p_K)$ of generating the different classes by *simultaneous confidence intervals* with a given confidence level of $1 - \alpha$. Here, $p_k$ represents the probability of generating the class of events $A_{\omega_k}$. From this specification, a procedure for constructing a possibility distribution is described, insuring that the resulting possibility distribution will dominate the true probability distribution. Since the probabilities $p$ of generating classes are unknown, we can build confidence intervals for each of them. In interval estimation, a scalar population parameter is typically estimated as a range of possible values, namely, a confidence interval, with a given confidence level $1 - \alpha$.

To build confidence intervals for multinomial proportions, it is possible to find simultaneous confidence intervals with a joint confidence level $1 - \alpha$. The method attempts to find a confidence region $\mathcal{C}_n$ in the parameter space $p = (p_1, \ldots, p_K) \in [0;1]^K | \sum_{i=1}^{K} p_i = 1$

as the Cartesian product of $K$ intervals $[p_1^-, p_1^+]...[p_K^-, p_K^+]$ such that we can estimate the coverage probability with:

$$\mathbb{P}(p \in C_n) \geq 1 - \alpha \tag{6}$$

At this moment, we can use the Goodman [28] formulation in a series of derivations to solve the problem of building the simultaneous confidence intervals. We define the quantity:

$$A = \chi^2(1 - \alpha/K, 1) + N \tag{7}$$

where $\chi^2(1 - \alpha/K, 1)$ denotes the quantile of order $1 - \alpha/K$ of the chi-square distribution with one degree of freedom, and $N = \sum_{i=1}^{K} n_i$ denotes the size of the sample. We also have the following quantities:

$$B_i = \chi^2(1 - \alpha/K, 1) + 2n_i, \tag{8}$$

$$C_i = \frac{n_i^2}{N}, \tag{9}$$

$$\Delta_i = B_i^2 - 4AC_i, \tag{10}$$

Finally, for each class of variability $A_{\omega_K}$, the bounds of the confidence intervals are defined as follows:

$$[p_i^-, p_i^+] = \left[ \frac{B_i - \Delta_i^{\frac{1}{2}}}{2A}, \frac{B_i + \Delta_i^{\frac{1}{2}}}{2A} \right] \tag{11}$$

It is now possible, based on these above interval-valued probabilities, to compute the most likely distributions of a class dominating any particular probability measure. Let $P$ denotes the partial order induced by the intervals $[p_i] = [p_i^-, p_i^+]$:

$$(\omega_i, \omega_j) \in P \Leftrightarrow p_i^+ < p_j^- \tag{12}$$

This partial order may be represented by the set of its compatible linear extensions $\Lambda(P) = \{l_u, u = 1, L\}$ or, equivalently, by the set of the corresponding permutations $\{\sigma_u, u = 1, L\}$. Then, for each possible permutation $\sigma_u$ associated to each linear order in $\Lambda(P)$, and each class $A_{\omega_i}$, we can solve the following linear program:

$$\pi_i^{\sigma_u} = \max_{p_1,...,p_K} \sum_{\{j | \sigma_u^{-1}(j) \leq \sigma_u^{-1}(i)\}} p_j \tag{13}$$

under the constraints:

$$\begin{cases} \sum_{i=1}^{K} p_i = 1 \\ p_k^- \leq p_k \leq p_k^+ \qquad \forall k \in \{1, \ldots, K\} \\ p_{\sigma_u(1)} \leq p_{\sigma_u(2)} \leq \cdots \leq p_{\sigma_u(K)} \end{cases} \tag{14}$$

Finally, we can compute the possibility degree of the cluster $A_{\omega_k}$ dominating all the distributions $\pi^{\sigma_u}$:

$$\pi_i = \max_{u=1,L} \pi_i^{\sigma_u} \qquad \forall i \in \{1, \ldots, K\} \tag{15}$$

### 6.2. Inferring Possibility Distribution for MIVs Inside an MAV

After clustering with a GMM model, we can calculate the posterior probability distribution of the datasample [29]. For a cluster $k$ and a data point $x_t$, the posterior probability is derived as ($\theta$ is the vector of parameters of the GMM model):

$$\Pr(k|x_t, \theta) = \frac{\pi_k \phi(x_t|\mu_k, \sum_k)}{\sum_{n=1}^{K} \pi_n \phi(x_t|\mu_n, \sum_n)}. \tag{16}$$

In addition, transforming a probability measure into a possibilistic one then amounts to choosing a possibility measure in the set $\Im(P)$ of possibility measures dominating $P$. This should be carried out by adding a strong order-preservation constraint, which ensures the preservation of the shape of the distribution:

$$p_i < p_j \Leftrightarrow \pi_i < \pi_j \qquad \forall i, j \in \{1, \ldots, q\}, \tag{17}$$

where $p_i = P(\{E_{\omega_i}\})$ and $\pi_i = \Pi(\{E_{\omega_i}\})$, $\forall i \in \{1, \ldots, q\}$. It is possible to search for the most specific possibility distribution verifying (5) and (17). A possibility distribution $\pi$ is more specific than $\pi'$ if $\pi \leq \pi'$, $\forall i$. The solution of this problem exists, is unique and can be described as follows. One can define a strict partial order $P$ on $\Omega$ represented by a set of compatible linear extensions $\Lambda(P) = \{l_u, u = 1, L\}$. To each possible linear order $l_u$, one can associate a permutation $\sigma_u$ of the set $\{1, \ldots, q\}$ such that:

$$\sigma_u(i) < \sigma_u(j) \Leftrightarrow (\omega_{\sigma_u(i)}, \omega_{\sigma_u(j)}) \in l_u, \tag{18}$$

The most specific possibility distribution compatible with the probability distribution $p = (p_1, p_2, \ldots, p_q)$ can then be obtained by taking the maximum over all possible permutations:

$$\pi_i = \max_{u=1,L} \sum_{\{j|\sigma_u^{-1}(j) \leq \sigma_u^{-1}(i)\}} p_j \tag{19}$$

The permutation $\sigma$ is a bijection and the reverse transformation $\sigma^{-1}$ gives the rank of each $p_i$ in the list of the probabilities sorted in ascending order. The number of permutations ($L$) depends on the duplicated $p_i$ in $p$. It is equal to 1 if there is no duplicate $p_i$, $\forall i$ and, in this case, $P$ is a *strict linear order* on $\Omega$.

Complexity

The complexity of our computational procedure is related to the discovery of the possibility degrees of the different $K$ classes. To solve this problem, the conceptually simplest approach is to generate all the linear extensions compatible with the partial order induced by the probability intervals, and then to solve the associated linear programs (i.e., Equation (19)) . However, this approach is, unfortunately, limited to small values of $K$ (say $K < 10$) due to the complexity of the algorithms generating linear extensions of complexity $O(L)$, where $L$ is the number of linear extensions. Even for moderate values of $K$, $L$ can be very large ($K!$ in the worst case) and generating all the linear extensions and solving the linear programs soon becomes intractable. A new formulation of the solution can be derived to considerably reduce the computations. This formulation is based on several steps. First, all the linear programs to be solved will be grouped in different subsets; then, an analytic expression for the best solution in each subset will be given; and lastly, it will be shown that it is not necessary to evaluate the solution for every subset. A simple computational algorithm will be derived, see [27] for more details. In addition, the actual complexity might be close to $O(|P_i|)$, where $P_i$ denotes the set of indices of the classes with a rank possibly, but not necessarily smaller than $\omega_i$.

## 7. Validation and Results

### 7.1. Experimental Dataset

To validate our approach, we use a three-year collection of global solar-radiation data recorded by the LaRGE laboratory facility, from 2011 to 2013. This dataset is measured in 1-second timestamps. We performed some aggregation by resampling the data to different timestamps, from 5 min to 60 min increments, to make our technique more robust when performing with several types of granularity. The method presented here deals with global horizontal irradiance (GHI), which is applicable to solar PV systems. Before using the data, it is important to remove the clear sky index, since it is a deterministic process. Within the normalization procedure, we obtain a signal without the deterministic component and we form several samples at different time horizons including 1, 5, 10, 30 and 60 min, upon which we apply the whole procedure.

The solar irradiation is normalized by the theoretical clear sky, $\text{GHI}_{csk}$ curve. The global horizontal irradiance (GHI) is the total amount of shortwave radiation received by a horizontal surface on the ground, which consists of the direct irradiance and the diffuse irradiance. The $\text{GHI}_{csk}$ is the GHI calculated in the condition of clear sky, using the Kasten clear-sky model. This model accounts for atmospheric turbidity and solar elevation angle. The inputs to this model are air mass, Linke turbidity, and elevation [30]. The Linke turbidity factor is a very convenient approximation to model the atmospheric absorption and scattering of the solar radiation under clear skies. It describes the optical thickness of the atmosphere due to water vapor and the aerosol particles relative to a dry and clean atmosphere. With larger Linke turbidity, there is more attenuation of the radiation by the clear-sky atmosphere. We obtain, then, the clear-sky index, $k_c$, defined as:

$$k_c(t) = \frac{\text{GHI}(t)}{\text{GHI}_{csk}(t)} \tag{20}$$

The input data at time ($t$) is then normalized into $\bar{P}(t)$, the normalized value of the solar irradiance with respect to the maximum value at time ($t$), $P_{\max}(t)$.

$$\bar{P}(t) = \frac{P(t)}{P_{\max}(t)} \tag{21}$$

This maximum value can be retrieved from the $\text{GHI}_{\max}$ curve with the following equation:

$$P_{\max}(t) = \frac{\text{GHI}_{csk}(t)}{\max(\text{GHI}_{csk})} \text{PV}_{\text{installed}} \tag{22}$$

### 7.2. Tracking Microscopic Variabilities—MIV

A careful analysis of the MAVs shows that the Viterbi decoder is unable to detect all the sets of time intervals where we observe the same meteorological conditions. From the perspective of trying to understand all the meteorological conditions that might have a significant influence on the solar irradiance, one should analyze the macroscopic fluctuations to identify the abrupt changes. We present a new formalism based on possibility distribution to detect these special variations. In Section 6, we developed a procedure of calculating the degrees of possibility of the dataset with the contribution of the posterior probability distributions of these data inside the different clusters or MAVs. The possibility degrees dominate the true probability distribution, as we can see in Figure 2. The degree of possibility clearly shows that the classes that might appear more frequently should have a high degree and the rare and specific phenomenon seen as drastic abrupt changes should be given a low degree of possibility. The results drawn in Figure 3a–c confirm this hypothesis and show clearly how the MIVs buried in the MAVs are detected. If one performs a careful visual inspection of these results, he/she sees that the types of fluctuations covered have a great and significant deviation in their amplitude regarding the rest of the sub-sequence.

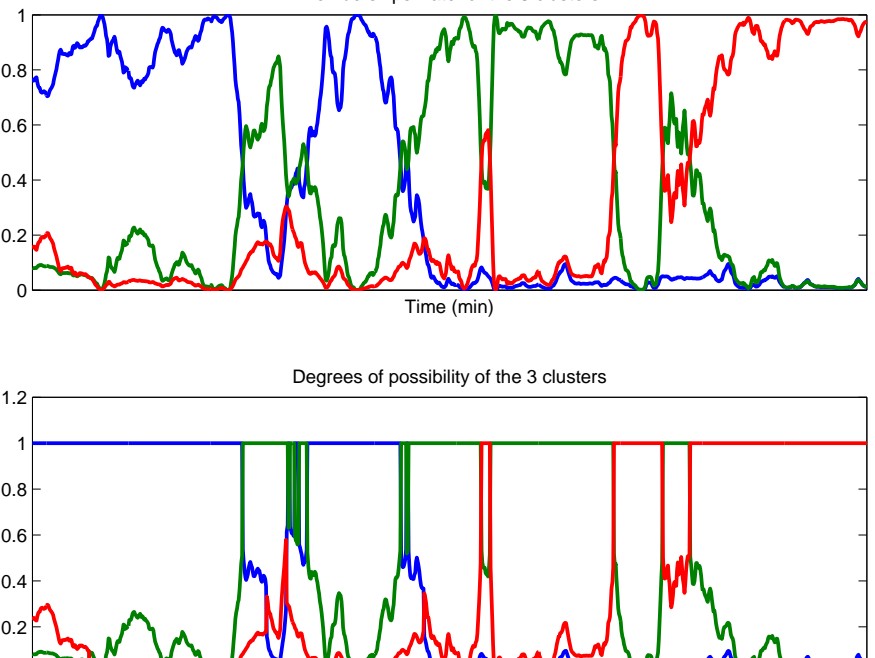

**Figure 2.** Possibility distributions of the dataset for the three states which dominate the probability distributions. Cases where the dataset are resampled in 10 min bins.

Another result of this analysis concerns the importance of the probability of the different MAVs. In probability theory, one generally asked question is to quantify a variable by a probabilistic density function. This problem does not have always a solution. The theory of possibility is another way to solve this problem, as it offers the possibility of affecting a value to a given variable to estimate its existence. By applying Equation (15), we can set a degree of possibility to each state. States with high degrees of possibility are most likely to happen for long time scales.

In our experience, we discovered a result which is not obviously a-priori. For each state $j$, we calculate the percentage of its duration with the formula $\left[ P \sum_{i=1}^{N^j} \left( S^j(i) \right) \right] / D$, where $P$ is the increment of the sampling period, $S^j$ the vector containing the sizes of the $N^j$ sub-sequences found by the Viterbi decoder and $D$ the total duration of all the states. We found that the rate of state duration is strongly correlated with the degree of possibility of the states. The two quantities vary in the same direction, as we can see in Table 1. This result is not at all obvious since the degree of possibility of a state is built using the proportions (in term of probability) of the other states/classes. Therefore, the result clearly shows that the state with the highest degree of possibility also has the longest duration.

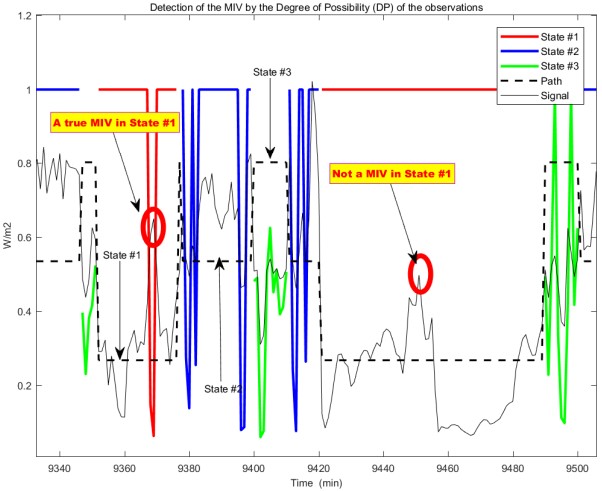

(**a**) Detection of MIVs, Time (10 min bins)

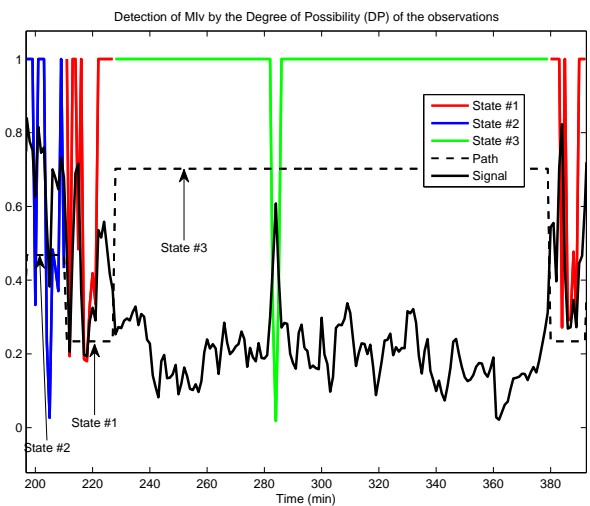

(**b**) Detection of MIVs, Time (5 min bins)

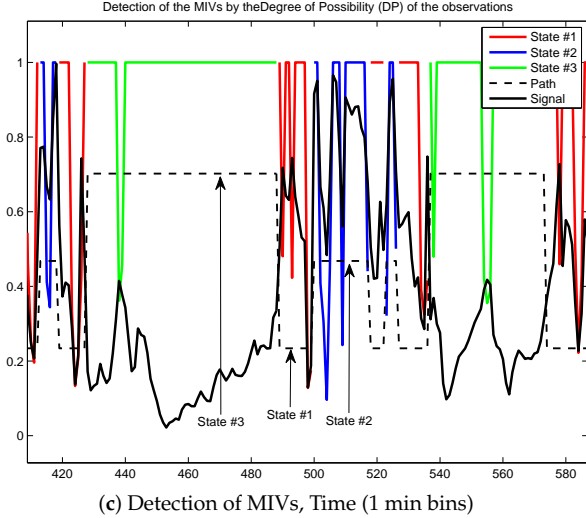

(**c**) Detection of MIVs, Time (1 min bins)

**Figure 3.** Detection of the MIVs by the degrees of Possibility (in color red, blue and green) of the dataset. Each HMM state detects the pics (meaningful abrupt changes) of the corresponding Viterbi path (sub-sequence). The Viterbi path has three levels corresponding to the three HMM states of variability. The MIVs correspond to bins where the degree of possibility is minimal. Case where the dataset is, respectively, resampled in 10, 5 and 1 min bins.

**Table 1.** Correlation between the state degree of possibility and the proportion of state duration. $p_i^-$ and $p_i^+$ are the lower and upper bounds of the confidence interval of the probability distribution of the state, where the true probability of the state is located and $\pi_i^S$ is the corresponding degree of possibility.

| Sampling period = 1 s | | | |
|---|---|---|---|
| State $i$ | 1 | 2 | 3 |
| $p_i^-$ | 0.2599 | 0.5484 | 0.1751 |
| $p_i^+$ | 0.2710 | 0.5610 | 0.1848 |
| $\pi_i^S$ | 0.4516 | 1.0000 | 0.1848 |
| Duration Rate | 37% | 61% | 2% |
| $\pi_{regular}$ | | 0.3821 | |
| Sampling period = 5 min | | | |
| State $i$ | 1 | 2 | 3 |
| $p_i^-$ | 0.2285 | 0.2855 | 0.3921 |
| $p_i^+$ | 0.2888 | 0.3496 | 0.4603 |
| $\pi_i^S$ | 0.5776 | 0.6079 | 1.0000 |
| Duration Rate | 2% | 38% | 60% |
| $\pi_{regular}$ | | 0.5917 | |
| Sampling period = 10 min | | | |
| State $i$ | 1 | 2 | 3 |
| $p_i^-$ | 0.1410 | 0.2478 | 0.4862 |
| $p_i^+$ | 0.2152 | 0.3362 | 0.5832 |
| $\pi_i^S$ | 0.2152 | 0.5138 | 1.0000 |
| Duration Rate | 2% | 37% | 61% |
| $\pi_{regular}$ | | 0.4806 | |

*7.3. Discovering the Probabilistic Distributions of the Parameters*

We recall that this study is relative to the prediction of solar irradiance, a very important element for the implementation of an appropriate tool for the prediction of energy of solar and photovoltaic origin in a geographical area whose meteorological conditions vary significantly. Generally, a forecasting framework is deterministic, parametric or non-parametric. When the useful information about the probabilistic laws which govern the intrinsic process of the studied system can be properly defined, parametric models can be used. Therefore, here, we investigate an interesting issue related to the identification of the probability distributions of the characteristic parameters of our different MAVs. We believe that the three kinds of MAVs we extracted with the Viterbi decoder might correspond to different statistical models. In addition, if we know exact information about each of these models, we could build PV forecasting techniques for each and combine them into a single prediction tool for the end manager.

In this work, we discover three kinds of macroscopic variation corresponding to the three HMM states. If we carefully inspect these three types of MAVs, we see that one of them has a more regular trend than the two others. Therefore, it is important to distinguish this regular type of MAV, since it might correspond to the situation where the atmospheric conditions are more likely to happen. The two other states will be the ones where these conditions change suddenly. Using the model-based possibility distributions, we found a decision method to extract and automatically label the regular state. A class will be set as "regular" if its degree of possibility is less than a given threshold $\pi_{regular}$, which we extract by means of the degrees of possibility related to the data, as described in Equation (19). This equation calculates the possibility distribution of each data point $x_t$ of the sample $x$. We obtain a matrix $\pi_K^N$ of dimension $K \times N$ ($K$ is the number of components, i.e., clusters/MAVs) and $N$ is the length of the data sample $x$). We take the average (**mean**) for each column (each column contains the possibility distribution for data point $x_t$) lying

in all clusters. Then, we obtain a second matrix $\pi_1^N$ and, finally, we use the following equation to derive the decision variable $\pi_{regular}$:

$$\pi_{regular} = \max(\pi_1^N) \tag{23}$$

From the obtained results we give in Table 1, one can see much useful information. First, we observe that the class with the smallest degree of possibility is always considered as the "regular" sequence, which happens less frequently and where the fluctuations are neither highly increasing nor highly decreasing. In addition, in this class, we also have the fewest number of observations. This situation is particularly interesting and it corresponds to a physical aspect of the behavior of the solar irradiance. Since the meteorological conditions are very likely to change in Guadeloupe, the fluctuations observed in the irradiance are most likely changing. The results also reveal that the "regular" state happens less frequently with the lowest duration rate; the other cases (highly increasing and decreasing) are observed more often.

Another result is related to the underlying probability distributions of the sequences of data under the different classes of MAVs. We would like to know if the three classes of fluctuations have the same laws. As we propose a parametric model, we also want to determine the parameters of these models in order to be able to implement them and to be able to decide when it is appropriate to completely recalibrate the global system. In other words, could we find some parameters describing the behavior of each class of MAV and establish parametric distributions for them? To address this issue, we proceed in several steps.

We consider the data in each class of MAV and calculate some statistical parameters as the mean, the standard deviation and the coefficient of variation. These features suffice to characterize and differentiate the statistical properties of the different MAVs. Their choice is obvious, since we make the assumption that the underlying distribution of the processes under consideration is a mixture of Gaussians and a Gaussian is typically characterized by its first- and second-order moments.

Each MAV is a sequence of disjoint sub-sequences (of different lengths) interlaced in time with the other sub-sequences of the other MAVs. Thus, for an MAV, one of its sub-sequence might not have enough data to calculate the needed statistical parameters. Therefore, to ensure that we can always have these given features, we sub-divide each MAV into a set of sub-sequences of suitable length. The best time window (we defined as TW) to subdivide an MAV is taken as the length of the sub-sequence with the highest duration rate. We take the sequence of maximum size because it is the one which contains, statistically, all the information needed to detect an MAV category. In this present work, we focus on the extraction of the underlying statistical properties of the MAVs. Another scope of this sub-division is to make it possible to apply, for our future PV forecasting technique, (re)-calibration for periodic time intervals in each MAV, to ensure that the model will have enough data to set the best values of the parameters.

The application of this subdivision procedure is not performed in the same way for the different types of MAV. Therefore, for the "regular" MAV, due to the regularity observed in the fluctuations, we divide the whole sequence into a set of sub-sequences of length TW.

For the "non-regular" MAVs, we might take into account the high variance in the data and the abrupt changes. Therefore, for each MAV, we form two other MAVs, named Type I and Type II, and, finally, each type is subdivided into several sub-sequences of length TW. Type I is the set of data corresponding only to the abrupt changes and the remaining data are labeled as Type II.

After performing the subdivision, we calculate the probabilistic distributions of the three parameters, namely, the mean, standard deviation and coefficient of variation, and, thereafter, we build the corresponding histograms. Finally, the data in the different cases are properly characterized by the two following parametric distributions, namely, a non-centered and non-normalized Gaussian probability distribution defined as:

$$y(x) = a \exp^{-(x-x_0)^2/2\sigma^2} \tag{24}$$

and a polynomial form of the probability distribution:

$$y(x) = a_0 + a_1 x + a_2 x^2 + a_3 x^3 + a_4 x^4 + a_5 x^5 + a_6 x^6 \tag{25}$$

In Figures 5–7, we draw the density probabilities, respectively, for the coefficient of variation, the mean and the standard deviation of Type-I and Type-II data sequences for states 1 and 3, when the sampling period is 1 min. The mean parameter can be modelled by the non-centered and non-normalized Gaussian probability, while the standard deviation and the coefficient of variation are modelled by the polynomial function with given parameters. For the sequence represented by the regular data, we see in Figure 4 that the mean (Figure 4a), STD (Figure 4b) and coefficient of variation (Figure 4c) are modelled by the non-centering Gaussian density or by the polynomial function. In this case, the choice to model the distribution of the process can be made either by the Gaussian density or the polynomial function. However, the non-centered Gaussian gives a better model than the polynomial.

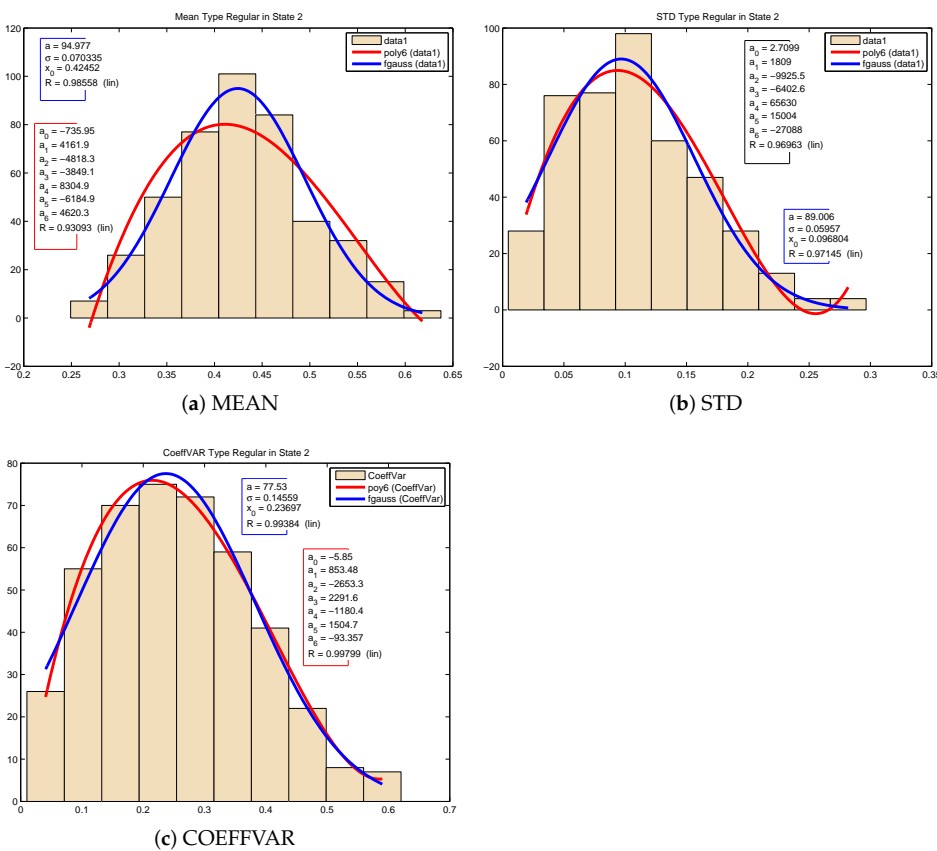

(**a**) MEAN        (**b**) STD

(**c**) COEFFVAR

**Figure 4.** Probability distribution of MEAN, STD and COEFFVAR parameters for data segmentation Type Regular for states 1 and 3. Case where the sampling period is 1 min bin increment.

The calculation of the parameters of the probability laws which govern the stochastic process of each MAV plays a dual role. First, it allows to define the appropriate model to apply to the future framework for the prediction of photovoltaic energy production. Then, these parameters constitute the means of identifying whether or not the system should be recalibrated to re-detect new MAVs if the atmospheric situation requires it. Indeed, if the limits of the confidence intervals for calculating these parameters go below a certain value, this would mean that the global model should be rebuilt to define a new one. For

the model to remain usable with high performance, one should ensure that the definition of the parameters is performed with a confidence interval of at least 95% (Figures 5–7).

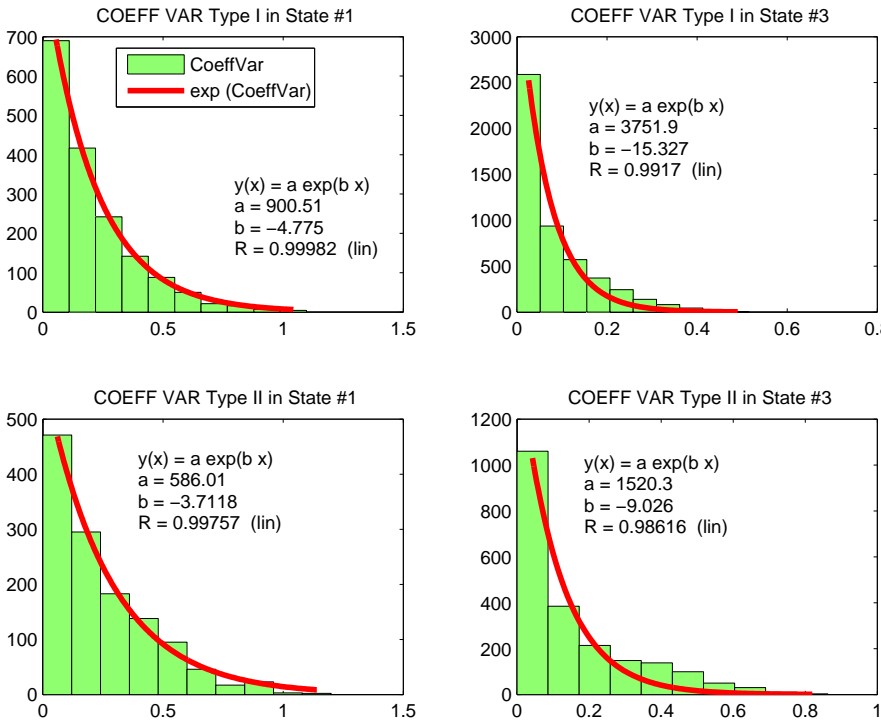

**Figure 5.** Probability distribution of COEFFVAR parameter for data segmentation Types I and II for states 1 and 3. Case where the sampling period is 1 min bin increment.

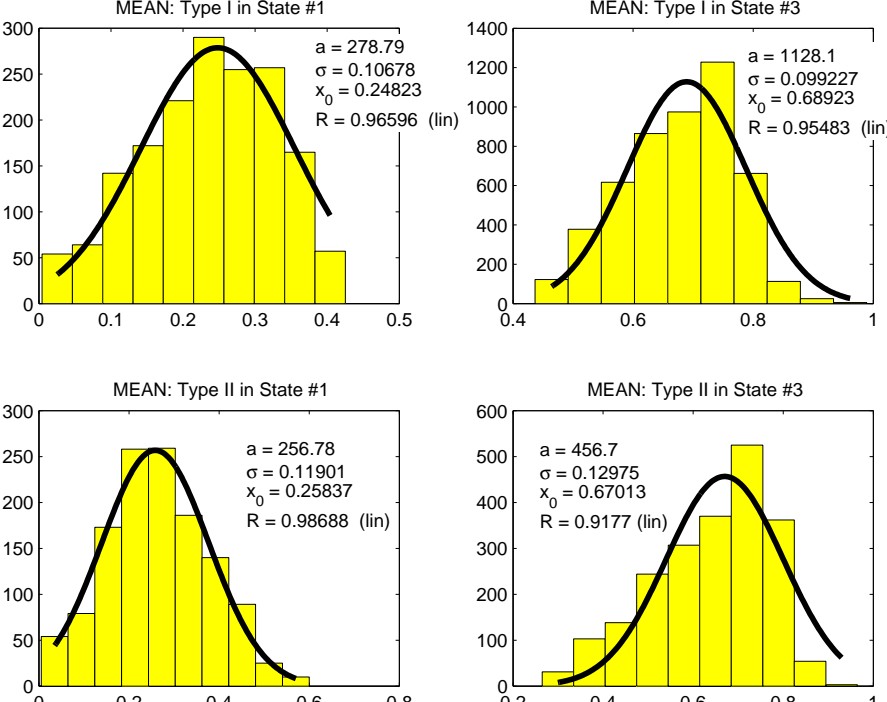

**Figure 6.** Probability distribution of MEAN parameter for data segmentation Types I and II for states 1 and 3. Case where the sampling period is 1 min bin increment.

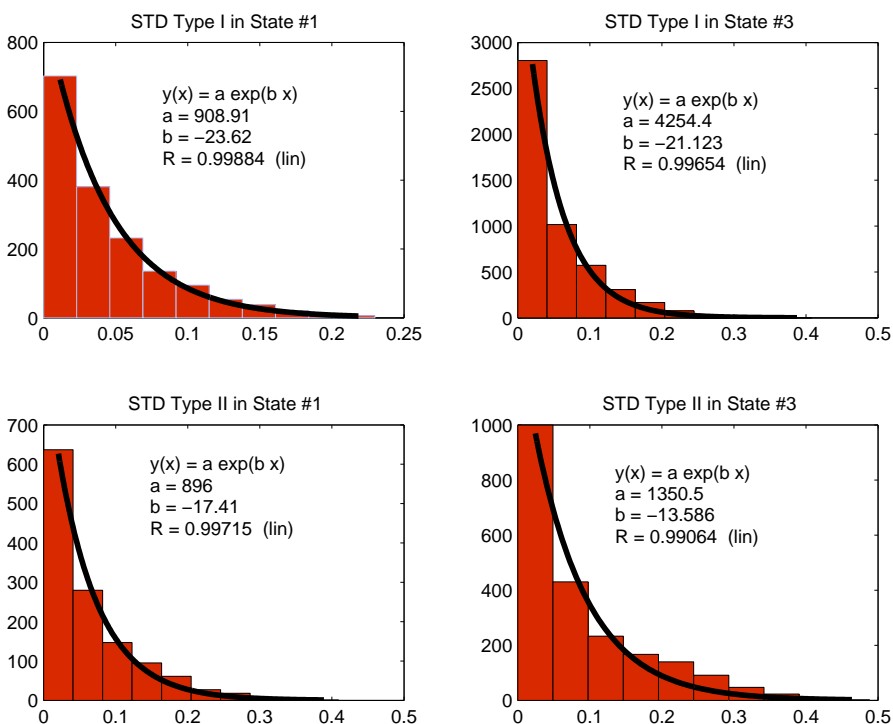

**Figure 7.** Probability distribution of STD parameter for data segmentation Types I and II for states 1 and 3. Case where the sampling period is 1 min bin increment.

## 8. How to Use the Extracted Knowledge about MAVs and MIVs to Build the Forecast Framework

If we want to set up an effective tool for the prediction of solar and photovoltaic energy, we must first control for the variability caused by atmospheric conditions. Fluctuations in solar irradiance, wind, cloud cover, amd temperature are all exogenous variables that can strongly influence the performance and accuracy of a prediction tool. The implementation of a good solar-irradiance prediction scheme should take into account two fundamental elements. First, it is necessary to be able to identify the time intervals where the atmospheric conditions are practically the same. This amounts to dividing the underlying process into different classes of statistical models, each with its own parameters. In our study, these statistical models were defined as macroscopic variables (MAV) and the paradigm of Gaussian mixture models allows us to identify them clearly, as described above. However, in certain regions of the globe where the atmospheric conditions are untimely and very changeable, it often happens that, for a short period of time, the model undergoes sudden deviations which can have severe consequences for the production of energy. Thus, to avoid performance drops, one should identify the instants of time when these abrupt changes take place and automatically recalibrate the system parameters. In addition, if the changes become too important during a given interval, the system should definitely choose another, more appropriate, model. To solve this problem, it is necessary to study the spatio-temporal correlations in the process. Therefore, as a second step, we resolve this problem by proposing a solution consisting of applying a hidden Markov model on the sequence of classes/models found by the GMM and applying the Viterbi algorithm. This phase makes it possible to identify how the different classes are spread out and how they evolve over time. At each instant of time, the system knows the model to be applied. Finally, using a method carefully developed from the theory of possibilities, we detect the microscopic variations (MIV variables) which correspond to the abrupt changes. In Figure 3a–c, we show, for different time scales (1 min, 5 min, 10 min), the decomposition of the process into different classes of variability and, in each class, the detection of abrupt changes. In a given class, if we do not observe sudden changes, the parameters of the model remain the same; otherwise, a recalibration is necessary.

A major question in the implementation of this tool is how to know if a given model must persist in two time intervals. To answer this question, we studied the probability laws of certain performance criteria such as the first- and second-order moments and the coefficient of variations. If, in time interval T1, these statistics follow probabilistic distributions with well-defined parameters and, in a later time interval T2, these parameters change in value in a significant way, this means that the system is in a position to modify its different states. Clearly, we affirm that in this case, the reloading process of the GMM and HMM models must resume to potentially detect other states which definitely correspond to new meteorological situations not previously observed.

## 9. Major Challenges for This Framework for Forecasting Solar Irradiance and Wind Speeds

The implementation of such a framework has some weak points. Indeed, the main challenge is to control the number of classes/models that the system must have over a fairly long time interval. The global warming observed today is causing strong changes in the climate in the different geographical areas of the world. Therefore, if the global environmental conditions vary to different degrees from one point to another in the same region, such a model may lack precision if the detection of classes of variability becomes very challenging. The prediction system must, therefore, take climate changes into account and, therefore, be capable of effecting fairly quickly a readjustment of the classes if the situation so requires. Another point of weakness can be linked to the slowness of the management system to recalibrate the parameters of a model when the latter undergoes spontaneous variations. This means that the algorithms put in place must take very little time to run to find the correct parameters of the model which are subject to turbulence.

## 10. Conclusions

This paper tackles the critical problem related to the forecast of global solar radiation fluctuations for the scope of building efficient PV-power production forecasting tools. Thus, in this work, we implemented a decision support tool to predict the effects of solar irradiance and atmospheric conditions. In conditions where climate change is notable, it is important to study the variability in sunshine if we want to set up an efficient infrastructure for the forecasting of photovoltaic energy. The elaboration of such a framework was made possible with a succinct combination of the theory of probabilities and the theory of possibilities. The proposed model is based on the fact that two essential elements must be analyzed to correctly predict the variability in solar irradiance. The first element concerns the variability observable over a given time interval corresponding to particular meteorological conditions. The latter, which we labelled MAVs, could be detected in the form of classes corresponding to Gaussian probability laws, which we analyzed using the probabilistic models GMM and HMM. The detection of these classes is effectively carried out by the GMM model and their evolution on the timescale was calculated by the HMM model. The second element in the analysis of the variability in solar irradiance concerns the sudden changes that can take place within an MAV and which correspond to real-time measurements of atmospheric conditions due to wind, temperature, cloud cover, etc. To clearly and unequivocally identify these instants of time or MIV, we made use of possibility theory, which defines a mathematical framework which allows us to reinforce the tools offered by probability theory.

We also discussed an important point related to the MAVs themselves. In fact, among all the classes of detected MAVs, there is necessarily one which corresponds to the climatic conditions most generally observed, with less variability in the amplitude of the data, in the region/geographical area in question. This type of MAV was labeled the 'regular state', the others being transient states. It is, again, thanks to the theory of possibilities that we clearly distinguished these two categories of states.

We believe that this analysis should be a prerequisite which might serve as a framework which might be coupled with a framework to perform PV prediction in order to help

energy providers carefully control and manage their industry. Thereafter, we can extract meaningful sequences of states and study their laws by extracting the density probabilities of the main characteristics of the process, namely, the mean, the standard deviation and the coefficient of variation. Having this knowledge at hand, the model parameter values of the prediction tool could be instantly adjusted when the dynamics of the model change from one time step to another. In addition, knowing that the model can evolve over time to rediscover new states, i.e., classes of variability, we, therefore, proceeded to analyze the probability distributions of certain statistical parameters (mean, coefficient of variation and standard deviation) of each MAV in the purpose of defining the hypotheses for changing the model. Clearly, this means an MAV becomes obsolete when the parameters of the distribution laws are no longer correct according to the confidence intervals theoretically established for them.

We believe that this work could serve as a promising tool in the machine-learning field, to perform predictions for solar- and PV-power production. Indeed, instead of applying a prediction model with fixed input parameters, this model can, rather, be calibrated according to the weather situation. For each class of MAV, a typical scenario of parameter model settings can be set and, more importantly, inside an MAV, when an MIV occurs, the model could be readjusted immediately to have a better performance by reducing the error rates. We think that in situations where the weather conditions are typically variable, calibrating the system with various models and scenarios according to the variability in the solar flux could reduce the global prediction error rates and increase the accuracy of the prediction.

In this direction, we hope to reinforce the methodology with more materials. A first idea should be to reuse the theory of possibility with the results of the degree of possibility of the states, in order to classify each type of variability by a number in the interval $[0, 1]$. Clearly, it would be possible to see a state as "a class of high or low" fluctuations if its degree of possibility exceeds a certain given threshold. Moreover, the different types of variability would also correspond to some known parametric/non-parametric probability distributions with specific parameters. If we know the probability laws which govern the whole process of each type of MAV, that will help to identify where calibration and re-calibration should be properly performed. In addition, in that case, it would be possible to build generative simulation models to describe more general situations in the domain of PV production monitoring. We hope to explore these ideas soon.

**Author Contributions:** Investigation, T.S.; Writing—review & editing, J.N. All authors have read and agreed to the published version of the manuscript.

**Funding:** This research received no external funding.

**Institutional Review Board Statement:** Not applicable.

**Informed Consent Statement:** Not applicable.

**Data Availability Statement:** The data that support the findings of this study are available from the authors upon reasonable request.

**Acknowledgments:** This study was supported by the Laboratory LARGE of the University of the Antilles in Guadeloupe.

**Conflicts of Interest:** The authors declare no conflict of interest.

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
