# Peer review of "Extracting Statistical Properties of Solar and Photovoltaic Power Production for the Scope of Building a Sophisticated Forecasting Framework"

_forecasting, doi:10.3390/forecast5010001_

Round 1

Reviewer 1 Report

The paper presents the results of theoretical work, with the nature of basic research.

In the content of the article, the authors do not subject the proposed method to evaluation by others.

The concept of real-time radiation forecasting seems to be a concept that is not understood from the point of view of usability.

The calculations are carried out for data that is not verifiable because, aside from the location, the authors do not provide details to make it clear what radiation data is being analyzed. In addition, concise time slices of 400-150 minutes were evaluated from the point of view that the radiation changes are dependent on year-round parameters related to earth movement.

There are general statements in the paper without indicating specific results, such as in terms of the dominant state shown in the graphs.

The work also contains editorial errors:

- figures and tables are placed much later than references to them, making verification difficult,

- line 136 and 156 lack literature sources,

- axes of the graphs do have different descriptions or lack thereof.

Author Response

Dear Reviewer, thanks for your comments and suggestions which made the article better.

We have now complete the requested modifications and corections, english was also improved.

If the article is published other authors will be free to evaluate the proposed methodology; Since this approach is a very new one, developped by a mathematician and applied to the global solar radiation forecast by a physicist, it will be fully evaluated by these community.

 We have been working on forecasting on it's application for many years now, with significant publications in this field, we do understand this concept. We want here to go further the simple use of a forecast model for a given location and for all weather conditions. in the article A benchmarking of machine learning techniques for solar radiation forecasting in an insular context, we have evaluated different model and their limitation regarding weather conditions and related fluctuations over time and amplitude;

In Statistical parameters as a means to a priori assess the accuracy of solar forecasting models, we have related these fluctutions with different parameters and applied it to the selection of models regarding the fluctuations;

In this work we set up a methodology to adjust the models parameters regarding the observed fluctuations;

We have given the details for the used data: they are global solar radiation, recorded in guadeloupe with a 1 Hz frequency;

The concise time slice are used for illustrations and better comprehension, the methodology was applied on a complete year of data;

Thanks again for your comments.

Reviewer 2 Report

This work is very interesting, and information about the weather situation could help improving several forecasting methods. I have some comments about this work:

- Open access to code and data might help both for replicability and implementation for other works.

- English needs reviewing.

- Introduction should have some references to justify why the methodology in this work is proposed and the improvements it offers compared to those works.

- Previous work only references one work from the authors. Should I think there should be better blibliography review.

- Work should point out the advantages of the proposed methodology compared with other techniques such as  PCA or wavelet decomposition.

- Figure 3,4 and 5 are distorted. Figure 5 has no label for time axis.

- Overall quality of several figures should be improved to have similar text sizes, labels on the axis, etc.

- Some references are showing [?].

- Would be interesting to compare results for different locations and climatologies to see the performance of the proposed methodology.

- Since authors mention the 10% RMSE on their previous forecasting work [1], it would be good to show how much does the proposed technique improve the previous forecasting results. This would also prove the point about improving forecasts.

Author Response

Dear Reviewer, thanks for your comments and corrections, making thus the article better;

We consider giving access to data and codes to the community upon request. We have made english corrections and added some more references and bibliography;

The explanations where improved for better understanding.

Thanks again.

Round 2

Reviewer 2 Report

- The overall quality of the manuscript has improved and is more clear.

- I would recommend some sort of comparison of the method applied on different locations, since weather variations will be different. It could be interesting to compare the states obtained depending on the different climatologies.

- English has also improved. Some minor faults, like "whether" in lines 36 and 81 (should be weather).

- Figure 3 has scientific notation on the X axis for a 150 minute time span but the rest of figures use minutes for that axis.

-Some formatting problems: Figures 6, 7, 8 are placed partially outside of the document; Figure 4 takes a whole page.

Author Response

Dear Reviewer, 

thanks for your comments and corrections;

We plan to have a work to analyse and compare different locations variability;

In this paper the aim is to present the methodology, since this one is itself not very simple, it needs a reference paper to be fully explained and exposed; 

The minor faults in the paper have been corrected, in red in the documents.

Figures have also been corrected and the format change for some of them.

Best Regards
